# Advanced Practice Nursing in Cardiology: The Slovak Perspective for the Role Development and Implementation

**DOI:** 10.3390/ijerph18168543

**Published:** 2021-08-12

**Authors:** Beáta Grešš Halász, Lucia Dimunová, Ivana Rónayová, Viliam Knap, Ľubomíra Lizáková

**Affiliations:** 1Department of Nursing, Faculty of Medicine, Pavol Jozef Šafárik University in Košice, Tr. SNP 1, 040 01 Košice, Slovakia; lucia.dimunova@upjs.sk; 2East Slovakian Institute of Cardiovascular Diseases (VÚSCH), a. s., Clinic of Angiology- ICU, Ondavská 8, 040 11 Košice, Slovakia; ivana.ronayova@gmail.com; 3Department of Physiatry, Faculty of Medicine, Balneology and Medical Rehabilitation, Pavol Jozef Šafárik University in Košice, Tr. SNP 1, 040 01 Košice, Slovakia; viliam.knap@upjs.sk; 4Department of Nursing, Faculty of Health Care, University of Prešov, Partizánska 1, 080 01 Prešov, Slovakia; lubomira.lizakova@unipo.sk

**Keywords:** advanced-practice nursing, cardiology, specialty, perception

## Abstract

Background: Cardiovascular diseases (CVDs) are the number one cause of death globally. Most can be prevented by addressing behavioral risk factors, where advanced practice nurses- clinical specialists in cardiovascular nursing play a fundamental role. This modern and effective role is based on advanced activities, knowledge, skills, and experience in a specialized field, which can make a significant contribution to solving the problems of these civilization diseases. The aim of this work is to explore the self-perception of advanced-practice nurses (APNs) working in cardiology and vascular medicine departments within the context of advanced-practice nursing. Methods: This quantitative exploratory study included 103 APNs working in cardiology and vascular diseases departments of specialized hospitals in Slovakia. A validated instrument was used. Results: The overall perception was at the level of 68.01%. The highest-rated domain was the outcomes for patients/clients, and subdomains were meeting the needs, education of healthcare workers, and quality in relation to management. There was a significant difference found among hospitals with a better scoring of specialized institutions. Conclusion: There have been promising advances due to the current legislation in Slovakia defining APNs and specialists’ competencies. However, the practice in nursing for CVD patients remains fragmented, uncategorized and less valued by stakeholders and the public. According to the results, nurses have the potential and preparedness for this role in the context of their knowledge and skills in general. The Authors conclude that there is a need of such specialization of APNs in Slovakia.

## 1. Introduction

Almost 18 million people die from cardiovascular diseases (CVDs) each year, an estimated 31% of all deaths worldwide. Of these, 85% are due to heart attack and stroke [1]. Mortality in 2019 caused by CVDs was 462.4/100,000 inhabitants in Slovakia [2]. CVDs are the number one cause of death globally. Most cardiovascular diseases can be prevented by addressing behavioral risk factors. People with cardiovascular disease or those at high cardiovascular risk require early detection and management using counselling and medicines, as appropriate [1]. The “Global action plan for the prevention and control of noncommunicable diseases 2013–2020” aims to reduce the number of premature deaths from noncommunicable diseases (NCDs) by 25% by the year 2025. The main objectives focus on prioritizing prevention, reducing risk factors and increasing control through people-centered health care, promotion, support of high-quality research, and monitoring trends in NCD prevention and control.. It includes strengthening human resources for prevention and control of NCDs such as knowledge, skills and improving the motivation of the health workforce, as well as career track development for healthcare workers through strengthening postgraduate training in various professional disciplines including nursing and optimizing the scope of nurses’ practice to contribute to prevention and control [3]. This fact provides space for the active application of the role in clinical practice.

Advanced Practice Nursing (APN) is a modern and effective nursing role for patients, management and nurses themselves. Its concept is based on nursing activities, knowledge, skills, and experience, which arise from adequate education and practice related to performance requirements and competencies. It builds on basic nursing education, which is enhanced by advanced knowledge and characterized by additional competencies and responsibilities [4].

The first “product” of the evolution of APN was a nurse-specialist in the19th century; later a nurse-clinical specialist in 1960s. Extended patient care did not pose a threat in the context of the existing professional hierarchy. In the USA, four categories of APN are defined. One of these is a Clinical Nurse Specialist—CNS [5]. In the UK, CNS is one of the specialist categories of Advanced Nurse Practitioner (ANP) [6]. Ireland has two main categories, where CNS is hierarchically the lower category, compared to ANP [7]. The scope of practice is direct practice, expert coaching and guidance, consultation, research, collaboration, ethical decision-making skills, and leadership. The standards of practice are assessment, diagnosis, outcome identification, planning, implementation, and evaluation [5].

Nursing specialization in cardiovascular diseases involves interventions related to the prevention and treatment of cardiovascular diseases. In Central European countries, a CNS working in cardiovascular diseases records the patient’s/client’s medical history and symptoms, administers prescribed medications, performs diagnostic tests based on ordinations, and assists with various examinations [8]. The definitions of CNS and APN are stated in the Ministry of Health of Slovak Republic (MoH SR) Decree no. 95/2018 Coll.: par. 2 (CNS) “…a nurse who has acquired professional competence shall provide nursing care and specialized nursing care provided in nursing specialized fields…”; par. 3 (APN): “...a nurse who graduated from at least the 2nd university degree (master’s), preceded by the 1st university degree (bachelor) in nursing, with specialization, and at least 5-year experience in a particular specialization, or a nurse without a specialization with 8-years professional experience.” Competencies of a CNS are listed in par. 1 (competencies of a nurse), extended by indication and change of an i.v. line, and application of medicines based on the medical doctor’s indication. Competencies of an APN are as previous competencies (par. 1 and 2) extended by interventions of a nurse (autonomously, on the basis of a medical doctor’s indication, and in a cooperation with a medical doctor) and a nurse specialist (autonomously), and of an APN (autonomously) as follows: assessment of needs; responsibility of an individual care plan; indication of prevention, nursing interventions; biological material collection after consultation with a medical doctor; decision and responsibility for the intervention and application of the drug treatment in accordance with the treatment plan specified by the medical doctor; medical aids´ indication and prescription; indication of the primarily healing wound´s treatment; ensuring compliance with relevant hygiene-epidemiological procedures; deciding on patient´s bed placement; nursing-team management; checking and analyzing records in the nursing documentation; creation, revision and evaluation of nursing standards and maps, educational plans and application to nursing practice; monitoring and carrying out nursing research, utilizing results in nursing practice; introduction and evaluation of nursing care quality system; training of nursing students [9].

Many countries worldwide are educating and recognizing CNS in Cardiac-Vascular Nursing [10]. For the nurse category in Slovakia, currently, there are 11 specialization divisions (anesthesiology and intensive care; instrumentation in the operating room; intensive nursing care—for adults; in neonatology; in pediatrics; nursing care for dialysis patients; nursing care for adults—until 2018, it was divided into nursing care for adults in fields of surgery, and in fields of internal medicine; nursing care in psychiatry; and perfusiology). Slovak legislation does not define nursing specialization in cardiology. However, there are nurses practicing in cardiovascular medicine departments with an official designation as advanced-practice nurses specialized in related nursing divisions such as anesthesiology and intensive care, intensive nursing care for adults, intensive nursing care in neonatology, intensive nursing care in pediatrics, and nursing care for adults. The aim of the research was therefore to find out whether these nurses perceive their own practice in the departments of cardiology and vascular medicine at an advanced level of nursing.

## 2. Materials and Methods

### 2.1. Design

Based on the background, the aim of this work was to explore the self-perception of advanced-practice nurses (APNs) working in departments of cardiology and vascular medicine within the context of advanced-practice nursing. This quantitative exploratory study was carried out in 2018–2019.

### 2.2. Population and Sample

The population of advanced-practice nurses working in departments of cardiology and vascular medicine in participating Slovakian institutes/hospitals—Faculty Hospital, Cardiology Clinic (FN, KK), Central Slovak Institute of Cardiovascular Diseases (SÚSCH), and East Slovakian Institute of Cardiovascular Diseases (VÚSCH)—were contacted via post. The study population was randomly selected on the basis of inclusion and exclusion criteria (designation, years of practice), and it was presented with an invitation letter attached to the final instrument.

### 2.3. Data Collection Procedures

Authors used the validated, retranslated, modified questionnaire previously constructed and tested by Prof. Begley et al. in 2010 in their SCAPE study [7], with an approval. The instrument consisted of two main parts—demography, and the questionnaire with three domains: outcomes for patients/clients; outcomes for nurses and other health care workers; and outcomes for managements. The reliability of the questionnaire was tested. The result of Cronbach α = 0.915 represented a high internal consistency of responses to the final questionnaire. Research packs were distributed to nurse managers in each hospital who further presented these to the nurses. Participants had a choice of completion of the instrument on paper or online. The approximate time of completion was 20 min.

### 2.4. Data Analysis

Descriptive statistics were calculated, and relationships were tested using IBM SPSS 25.0 software (IBM, Armonk, NY, USA). 

### 2.5. Ethical Considerations

The ethical approval and distribution process was provided by each institution/hospital. The study population was informed about the purpose, procedure, compliance with confidentiality and anonymity within the study, and was asked to complete the instrument individually without the use of other sources. The completed instrument was an expression of consent to participate. The investigation conforms with the principles outlined in the Declaration of Helsinki (Br Med J 1964; ii 177).

## 3. Results

### 3.1. Demographics

A total of 103 randomly selected advanced-practice nurses working in cardiovascular disease departments participated in this research. Demographic results are listed in Table 1: the mean age of respondents was over 39 years, and years of practice almost 17 years; female respondents were represented to a much larger extent in comparison to male respondents, the three participating hospitals were represented almost equally, almost all respondents completed MSc/DN, and more than two-thirds completed a specialization in related fields of practice.

### 3.2. Domains and the Questionnaire

The three domains consisted of several items relevant to the specific advanced role in nursing. Using a Likert scale from 1–7, respondents indicated their perception of the impact they believe they have on the outcome. The first domain analyzed outcomes related to patients/clients, where the highest-rated subdomain was related to meeting patients’ needs by nurses, and the lowest patients’ involvement in decision-making. The quality of life in terms of physical comfort was the highest-rated item, and the family/carer quality of life was the lowest. The second domain pertained to outcomes related to nurses and other health care workers, of which the highest-rated subdomain was related to education and the lowest-rated subdomain was related to research and evidence-based practice. Achievement of new educational interventions for professionals was the highest-rated item in comparison to research activity level in clinical practice, which was rated the lowest. The third domain concerned outcomes related to management with the highest-rated subdomain of quality, and the lowest-rated subdomain was the effectivity of care. Safety of care was rated the highest while the lowest-rated item involved waiting times. In relation to items in this domain, the highest score was achieved in safety of care, and the lowest in waiting times. Comparing domains, the highest-rated domain was the “outcomes for patients/clients”, followed by the “outcomes for managements”, and the lowest-rated was the “outcomes for nurses and other health care workers”. The general result of the nurses’ perception was positive (Table 2). 

### 3.3. Correlations and Differences

We presumed significant relationships between the perception of advanced-practice nurses working in the area of cardiovascular diseases and years of practice, within institutions/hospitals, and having or not having completed a specialization in related fields. For set hypotheses testing, we computed Pearson’s correlation coefficient to find the relationship between variables, and a Student’s t-test and ANOVA test to measure the differences among items in relation to the perception variable. Two significant differences were found between the perception of nurses working in SÚSCH (M = 255.39; SD ± 40.49) and FN, KK (M = 202.47; SD ± 41.00) with the highest mean difference M = 52.93 (*p* = 0.000), and the SÚSCH (M = 255.39; SD ± 40.49) and VÚSCH (M = 214.18; SD ± 43.30) with the second highest mean difference M = 41.21 (*p* = 0.000). Non-significant difference was found between perception of nurses without specialization (M = 226.20; SD ± 48.07) and with specialization in related areas (M = 223.20; SD ± 46.80), where, surprisingly, the first group of nurses achieved higher scores (Table 3.). No other significant relationships were found.

## 4. Discussion

### 4.1. Reasons to Implement the Role of APN in Cardiology 

Cardiovascular diseases remain a global issue of the 21st century [1], and predictions of their growth requires an appropriate preparedness and response from health care systems worldwide. The impact of CVDs in our society is tremendous [11].

To prevent first heart attacks and strokes, it is necessary to focus on individuals’ health care interventions for those who have a high overall cardiovascular risk or a single risk factor. Early intervention is more cost-effective and has the potential to reduce cardiovascular events. This approach is feasible in primary care in a low-resource environment, including non-physician health care professionals [1].

APNs are an integral part of a multidisciplinary team and have an irreplaceable role in providing quality nursing care. They provide a complex and continual care to patients/clients [12]. At present, there is a consensus that modern nursing interacts with clinical practice, education, research, and the professional development of nursing practice itself. The role of an APN is a promising starting point that has long-term positive impacts on the improvement and availability of services for patients/clients in times of their wellness or illness. It includes individuals and communities ranging from primary to quaternary care and includes new models of care structures and practices [13]. The role can make a significant contribution to the prevention and management of individuals at risk of, or with CVDs. The results of this research show the potential and the preparedness of nurses for this role in the context of their knowledge and skills in general. In relation to the perception resulting from the data analysis, we conclude that the results are positive, but it can be considered whether the level achieved is sufficient. The highest-rated subdomains were related to meeting needs, education, and quality of care. On the contrary, the lowest means were found in areas related to involvement of the patient in decision-making, research and evidence-based practice, or effectivity of care. The official establishment and introduction of the APN designation into Slovak legislation resulted from the demand for differentiation of categories of nurses in Slovakia and thus the inclusion of master-educated and erudite nurses into a higher category. However, the practice has not changed much. Research and EBP are part of the APN’s tasks only to a minimum extent. Other low results are due to the lack of autonomy of nurses, which is again related to the traditional approach in practice. Moreover, there were deficiencies explored among facilities and specialization. The differences in the perception related to participating facilities were surprising as they are all specialized in cardiology. However, results confirmed that highly specialized institutions may contribute to higher perception levels. An interesting finding was that specialized nurses’ perception was lower than those without specialization. The reasons for this were partially expressed in answers of the open-ended question. The dominant issues were related to insufficient workload management, lack of nursing staff, dissatisfaction with reimbursement and physical and psychological burden on workload, in addition to the nursing itself being negatively perceived by the public and other health care professionals. These would require a more thorough examination and analysis not only of the reasons that led to them, but also of the content of the nurses’ practice, and the content of the training and its conditions related to specialization preparation.

### 4.2. Scope of Practice

In the US, depending on the scope of practice and role, APRNs function across the continuum of care, and provide care in acute and long-term care facilities, primary care clinics and outpatients. 

The APN in Cardiovascular Nursing is licensed to practice as a registered nurse in the state in which the nurse practices and is subject to the state’s legal constraints and regulations. Many states have additional requirements that the nurse must satisfy to practice or be recognized. APNs have a broad scope of general as well as specific knowledge that is a prerequisite to specialty competencies within the clinical judgment, advocacy and ethics, caring practices, collaboration, critical thinking, responsiveness to diverse populations, clinical inquiry, and facilitation of learning. With years of practice, the nurse will move from a novice to an expert clinician [10].

Advanced Practice Nurses and Midwifes in Ireland offer wellness, interventions to promote a healthy lifestyle to patients/clients, families, and communities in collaboration with other professionals, as amended by practice and guidance. The care is provided by autonomous and experienced nurses who are reliable and responsible for their practice and the services and interventions they provide [14]. Nurses are highly experienced in clinical practice, critical thinking decision-making, leading, research, with a masters’ degree relevant to the area of specialist practice (e.g., RANP in Cardiology). Registration is regulated by the Nursing and Midwifery Board of Ireland with a structured framework [7].

In UK, a new framework for APNs identified four key pillars of the role: clinical practice, management and leadership, education and training, and research. Pillars are applied in practice. Roles are well-defined by specific groups of patients, driven by guidelines and supplemented by diagnostic reasoning skills, investigations to attain a diagnosis, and clinical management plans. A nurse is well trained to perform specific interventions, educate other health professionals, get involved in research and audit, and keep up-to-date with EBP practice. The role is dynamic and has the potential to lead to a nurse consultant role. However, presently, there is no regulation of APN and a standardization of pay and practice banding in the UK [15].

Recently published results of a survey by Scordo and Weidner (2020) revealed that the vast majority of APRNs specializing in cardiology worked full time as direct patient care providers in hospital settings. Very few had administrative or research positions. The top three areas of practice of the respondents were general cardiology, interventional cardiology and heart failure. The primary responsibilities reported by the surveyed respondents were patient education, consultations, outpatient follow-ups, hospital rounds and admission assessments and data collection. A minority of respondents performed elective cardioversion, catheter placement and/or endotracheal intubation. Most of the respondents reported being able to practice to the full level of their education and training. Only a few reported that they never were able to practice to their full level. One of these barriers was state nurse practice acts and nursing board terms, departmental supervisors’ lack of support to APRNs and institutional regulations. Currently, only 11 are internationally recognized valued members of team-based cardiology care of 2971 APRNs [16].

Kwok et al. (2020) described an innovative nurse-led cardiology assessment service that has improved standards and cares for patients with acute cardiology conditions. They showed an effectiveness of the service, with the ability reduce the time taken to receive a specialist opinion, length of stay and costs to the health service [17].

### 4.3. The Global Future of the Specialty Role

The global burden of CVDs is high and continues to grow. Preventing CVDs is imperative. In most countries, nurses represent the largest group of healthcare workers. They can make a significant contribution to prevention and treatment of NCDs such as CVDs [18]. One of the important roles of an APN in cardiology is communication with the patient/client and his family, where nurses help them understand the illness and educate them about appropriate medical procedures and methods of treatment. The APN participates in consultations with healthcare professionals in determining the risk factors for cardiovascular failure. Based on the acquired knowledge, the APN plans procedures for maintaining a favorable condition of the patient/client after an operation or treatment. These plans include education on changing the patient’s/client’s health style including nutrition, smoking, exercise, and more [8]. In many countries, the professionalization of nursing is conditioned by a standard level of education. It is not ideal if educated nurses perform the technical-medical interventions of doctors instead of their own ones [19]. According to Maier and Aike (2016), to maximize workforce capacity, many countries have implemented task-shifting reforms. Reforms focus on elimination of regulatory and financial barriers, yet they are lengthy and controversial. Primarily, countries are reforming education. From a global perspective, the development of standardized definitions, minimum requirements for education and practice would facilitate more effective recognition procedures [20].

There are tendencies to compare APNs’ and medical doctors’ competencies. The research of Voogt-Prius et al. (2010) compared the clinical effectiveness of practice nurses acting as substitutes for GPs in CVD risk management in Dutch primary care. APNs achieved results equal to or better than GPs. These findings support the involvement of practice nurses in CVD risk management [21]. These are relevant results; however, it is important to understand that an APN in Cardiology is not a substitute for a medical doctor. Pillars of practice go beyond this, as APNs are leaders, experts, and consultants in their clinical practice [15,16,17,18,19]. Solving issues of individual competencies of nurses can contribute to a quality and recognized partnership between the nurse and the medical doctor. Therefore, it is necessary to precisely define the practice of the APN. Education and autonomy increase the professional self-confidence of nurses. Unfortunately, some medical doctors as well as other health professionals misunderstand advanced-practice in nursing. Time-consuming interventions, less demanding interventions, and some clinical interventions typically done by medical doctors are allocated to APNs believing that this is their role. This approach leads to conflicts between nurses and medical doctors. APNs must constantly defend their autonomy, competence and abilities [19,20,21,22,23]. Medical doctors and APNs work in collaboration. Multidisciplinary collaboration is a key point in the care and delivery of services to patients/clients, families and communities. In the context of advanced nursing practice, it is known that cooperation between nurses and medical doctors can increase the quality of care provided, patient/client satisfaction and, conversely, reduce mortality and morbidity. The implementation of advanced-practice can also bring a positive result in terms of the satisfaction of healthcare professionals themselves. However, interdisciplinary cooperation cannot take place until advanced nursing practice is recognized and valued by health systems and policies [22]. The role was created by natural evolution. At present, APN is integrated not only into the healthcare system, but also into the heart and soul of the nursing profession. It is a recognized, clinically innovative, controlled and regulated role with an adequate education system. There is a promising move forward due to the current legislation in Slovakia, defining APNs and specialists’ competencies. However, the practice remains fragmented and not categorized [19,20,21,22]. National and international organizations play an important role. All agreements concluded relate to conceptual models of advanced nursing practice among nurses, professionals, organizations, and politicians that aim to vector nursing to patients’/clients’ benefits. The future depends on the extent to which advanced nursing practice meets the needs of society, health systems, and public policies [13]. Based on our findings, several consequences have emerged:CVDs are the number one cause of death and APN in cardiology play an important role in care management.The results showed the readiness of APNs in Slovakia to this specialty where APNs working in specialized institutions scored better.There is a need for such specialization of APNs in Slovakia.

## 5. Conclusions

Given the current legislation in Slovakia defining the APN and the competencies of specialists, promising progress has been made. However, the nursing practice for CVD patients remains fragmented, uncategorized and less valued by stakeholders and the public. The results show that specialization of APN in Cardiology Nursing in Slovakia is necessary. Nurses have potential and preparedness for this role in the context of their knowledge and skills in general. The development and refinement of advanced education curricula, with a particular focus on research, evidence-based practice and its implementation, as well as advances in cardiology nursing, should bring significant benefits to nurses themselves, to managements, and particularly to the public and patients.

## Figures and Tables

**Table 1 ijerph-18-08543-t001:** Demography.

Item (*N* = 103)	Freq/%	M/SD±
**Age**	-	39.09/7.99
**Years of practice**	-	16.68/9.09
**Institution/Hospital**		
FN, KK	31/30.10	-
SÚSCH	33/32.04	-
VÚSCH	39/37.86	-
**Gender**		
Male	10/9.71	-
Female	93/90.29	-
**Highest achieved education**		
MSc/DN	101/98.06	-
PhD	2/1.94	-
**Specialization in related fields**		
Yes	73/78.87	-
No	30/29.13	-

**Table 2 ijerph-18-08543-t002:** Results of the questionnaire (domains and total).

Domains/Items	Min	Max	M	SD±	%
**Questionnaire Total**	**132.00**	**314.00**	**223.77**	**46.82**	**68.01**
**Domain 1: Outcomes for patients/clients**	**72.00**	**188.00**	**133.72**	**28.01**	**70.75**
Highest rated subdomain: meeting the needs	18.00	44.00	35.10	8.10	71.63
Lowest rated subdomain: involvement of the patient in decision-making	8.00	23.00	19.28	4.74	68.86
Highest rated item: quality of life in physical comfort	2.00	7.00	5.42	1.34	77.43
Lowest rated item: family/carer quality of life	1.00	7.00	4.36	1.43	62.29
**Domain 2: Outcomes for nurses and other health care workers**	**18.00**	**75.00**	**46.61**	**13.30**	**60.53**
Highest rated subdomain: education	7.00	28.00	18.35	5.16	65.53
Lowest rated subdomain: research and evidence-based practice	5.00	26.00	15.91	4.94	56.83
Highest rated item: achievement of new educational intervention (professionals)	1.00	7.00	4.69	1.60	67.00
Lowest rated item: research activity level in clinical practice	1.00	7.00	3.50	1.49	50.00
**Domain 3: Outcomes for managements**	**23.00**	**62.00**	**43.42**	**10.00**	**68.95**
Highest rated subdomain: quality	10.00	34.00	23.01	6.19	82.18
Lowest rated subdomain: effectivity	4.00	14.00	9.83	2.40	70.18
Highest rated item: safety of care	1.00	7.00	5.43	1.46	77.57
Lowest rated item: waiting times	1.00	7.00	4.28	1.62	61.14

**Table 3 ijerph-18-08543-t003:** Significant relationships among variables.

Relationships	Test	Test Results	Df	*p*
Perception vs. years of practice	Pearson Correlation	Corr. coef. = 0.151	n/a	0.130
Perception vs. hospital	ANOVA	F = 14.405	102	0.000 *
Perception vs. specialization	*t*- test	*t* = 0.295	99	0.769

Notes: * significance at *p* < 0.001.

## Data Availability

The data presented in this study are available on request from the corresponding author. The data are not publicly available due to General Data Protection Regulation (GDPR).

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
