# Peer review of "Advanced Practice Nursing in Cardiology: The Slovak Perspective for the Role Development and Implementation"

_ijerph, 2021, doi:10.3390/ijerph18168543_

Round 1

Reviewer 1 Report

It was with great interest that I read the paper devoted to Advanced Practice Nursing (APN) in cardiology. The APN might be the future of nursing, both in the context of various nursing fields and in relation to healthcare systems.

               While revising the publication, the following issues need to be addressed:

  • It would be beneficial to explain to the reader what types of nursing specialities exist in the Slovakian healthcare system and which of these specialities cover cardiac care.
  • It would be advisable to present inclusion and exclusion criteria adopted in this work.
  • The lines 149 -151 convey the same information. They should be rewritten.
  • The authors might want to elaborate on the questionnaire results related to the research and evidence-based practice. It might be useful to discuss why it has been the lowest rated subdomain. Such outcome may stem from the fact that majority of nurses do not have access to scientific publications, e.g. nursing journals.
  • Another point that might be worth looking into is the average age of cardiac care nurses, which is 39 years. In most European countries, and in the majority of nursing fields the median age of nurses is much higher, reaching 50 years.
  • There should be a clear separation between the discussion and the conclusion The summary which serves as a conclusion of the findings requires reediting and needs to refer to the research presented in the publication.

Reviewer 2 Report

This exploratory study is of interest as it pertains to the prevention and care of a common NCD by Advanced Practice Nurses in Slovakia.  It is important because it describes a role that is developing in a European country and would be applicable to other countries in the region. However, in order to augment a clear understanding of this study's findings, it will be necessary to more adequately define the primary variable that is termed "perception" or "perception level".  It is not clear what that means - is it perception of APNs or perception by APNs?  What are they perceiving?

Also, the description of APNs in the United States needs to be clarified.  It appears that the authors are conflating the roles of Clinical Nurse Specialists(CNS) and Nurse Practitioners (NP).  There are nurses that specialize in Cardiac nursing however, they may not be a CNS and may not be masters prepared.  Also, importantly, CNSs do not have ability to prescribe pharmacologic treatments.  Only NPs can diagnose and prescribe pharmacologic treatments.  Also, there are no CV NPs. NPs in the US specialize based on population focus and setting instead of disease - for example Adult-Gerontology Primary Care NP. This section would need to be re-written in order to clearly describe and compare  the role of interest. 
